# Production risk and technical efficiency of dry-season vegetable farmers in the Upper East Region of Ghana

**James Anaba Akolgo** [1]*, **Y. B. Osei-Asare** [2], **D. B. Sarpong** [2], **Freda E. Asem** [2], **Wilhemina Quaye** [3]

**1** Department of Ecological Agriculture, School of Agriculture, Bolgatanga Technical University, Bolgtanaga, Ghana, **2** Department of Agricultural Economics and Agribusiness, University of Ghana, Accra, Ghana, **3** CSIR-Science and Technology Policy Research Institute, Accra, Ghana

* akonabaj@gmail.com

**Data Availability Statement:** All relevant data are within the manuscript.

**Funding:** The author(s) received no specific funding for this work.

## Abstract

The Ghanaian population is aware of the increasing health challenges in our health facilities and the need to consume more vegetables to improve their health status. This, coupled with population growth and changing consumer patterns has led to an increasing demand for vegetable products in Ghana. Smallholder farmers in the country have thus intensified the production of vegetables during the dry season to meet consumers' demand and to generate income. However, their outputs have been lower than the country's potential, so the research was conducted to identify the causes and determinants of the low yields. A total of 322 dry-season vegetable farmers in seven (7) districts in twenty-four (24) communities were selected from the Upper East Region of Ghana using a purposive random sampling technique. The Kumbhakar model was employed to compute the production risk, technical inefficiency and determinants of vegetable production in the region. The study reveals that the input variables: labour, seed, fertilizer, agrochemical and irrigation costs positively are related to the output value of vegetables with an increasing return to scale. In addition, labour, seed and agrochemical costs show a significant production risk-decreasing effect while the risk of vegetable production is reduced with fertilizer and irrigation costs. The study further depicts that extension visits, experience, water pumps and gravity-fed irrigation systems positively affect the technical efficiency of dry-season vegetable production. Again, given the current state of technology and resources available to the farmers, enhancing the vegetable outputs could be achieved by reducing the technical inefficiencies by 27% while considering the effects of production risk. The study concludes that the farmers can improve the output of the vegetable farms for higher income by adopting the best vegetable production practices such as efficient water-saving irrigation technologies and fertilizer usage while adopting the knowledge from the extension training to improve their technical efficiency.

**Competing interests:** The authors declare that this article has no known potential conflict of interest (financial or non-financial) or personal relationships that could have appeared to influence the work reported in this paper.

## 1. Introduction

Although agriculture is the single largest contributor to most economies in the world, changing climatic conditions coupled with the recent COVID-19 pandemic have caused hardships, increased poverty and food insecurity among people [1]. In 2021 alone, the world's hungry population rose from 702 to 828 million [2]. Micronutrient deficiencies have also remained sticky from declining, affecting nearly 2 billion people [3]. West Africa including Ghana is projected to experience a 10–40% decline in crop yields and the growing season will shrink by 20% on average by 2050 due to climate change [4]. Given this backdrop, there is broad consensus globally that agriculture must double productivity while reducing its negative impact on the environment across scales for improved social outcomes. as contingent towards achieving the United Nations Sustainable Development Goals (SDGs), particularly on no poverty (goal 1), improved food security and nutrition (goal 2) and climate action (goal 13) [5,6].

Irrigated vegetable production is one promising agriculture sub-sector for sustainable income for smallholder farmers, government revenue and foreign exchange earnings while at the same time providing the needed micronutrients for the rural population [7]. Additionally, the value chain in vegetable production creates employment opportunities, helps to alleviate poverty and improves rural livelihood. Global statistics show that vegetable production grew remarkably, reaching about 1154 million tonnes in 2021, from 232 million tonnes in 1972 [8,9]. China is the leader in production with an impressive output of over 600 million metric tonnes, closely followed by India with approximately 140 million metric tonnes in 2021 [10,11].

Several studies have revealed a positive relationship between the consumption of vegetables and the global reduction of Non-Communicable Diseases (NCDs) [12–15]. The nutrients in vegetable products contain biochemicals, polyphenols and phytochemical compounds, minerals, fibre and antioxidants such as vitamins A, C and E which are important in neutralizing free radicals that cause a wide range of diseases [16,17]. Vegetables support good health and build up the immune system, which is critical in the fight against many cardiovascular diseases (e.g., heart attacks, stroke), cancers, chronic respiratory diseases (e.g., chronic obstructive pulmonary disease, asthma) and diabetes [18].

A major risk factor for most NCDs is an unhealthy diet, which includes low consumption of vegetables [18,19]. Multiple studies conclude that non-communicable diseases are responsible for approximately 70% of global deaths, and 77% of deaths in low and middle-income countries [19–21]. NCDs are prevalent in adults and are projected to increase substantially in the coming years due to the increasing ageing population occurring globally, particularly in low-income economies. The United Nations (UN) estimates that two-thirds of the world's population aged 60 years and over will live by 2050 [22], which will cause an increase in demand for vegetable products.

A study of 162 countries in the world showed that the weighted mean vegetable consumption was 186 g/day in 2020 [23]. The consumption in Africa ranges between 27 and 114 kg per capita/year but in sub-Saharan Africa (SSA) is much lower as less than a third of the 48 countries in the region consume sufficient vegetables to meet the World Health Organization (WHO) recommended minimum dietary requirement of 240g per person per day [23,24]. Currently, Ghanaians consume on average 2.3% of the WHO requirement of 4–6% vegetable portions in a diet per day [25]. Lower consumption of vegetables is considered a risk factor for most NCDs and malnutrition [26]. Given the importance of vegetables to the Ghanaian economy, concerns about raising income and ensuring food security for the citizenry should focus more on promoting vegetable production, however, the emphasis of the agricultural sector is always on how to increase the output of energy-rich staple crops like cereals, tubers and pulses [27,28].

This notwithstanding, vegetable production in Ghana has shown an upward trend since 1972 with the 2021 output of 788,693 tonnes representing a peak in production [29]. The trend in output growth, however, is primarily attributed to area expansion. The low yield of vegetables is due to inefficiencies in production and the effects of climate change [25]. The world temperature for instance has reached an alarming proportion in the last decade, which has led to a continued rise in sea level and erratic rainfall patterns, causing extreme weather variability and climatic events such as floods, drought, windstorms and heatwaves in recent years [30,31]. In Ghana, rising temperatures, shrinking annual rainfall duration and increased evapotranspiration now permit only 3 to 4 months of agricultural activities and 8 to 9 months of a prolonged dry season in northern Ghana [32]. This phenomenon disrupts farming activities, resulting in low regional food production which causes food insecurity [33,34].

As a climate change mitigation strategy, smallholder farmers in the Upper East Region of Ghana have resorted to commercial vegetable production to raise income to support their farm families. These farmers employ both conventional and non-conventional irrigation technologies to produce vegetables such as chilli, tomatoes, onions, and garden eggs during the dry season to enhance their livelihoods, however, their yield remains low [35]. The attainable yields of vegetables in Ghana are generally lower compared to neighbours like Niger, Burkina Faso, and Mali although with similar geographical conditions. For example, while farmers in Niger achieve about 28.1Mt/ha of onions, their Ghanaian counterparts harvest less than 11.0Mt/ha [35], which is less than 50% of the national mean yield. Similarly, while Egypt and Niger make 17Mt/ha and 14Mt.ha of chilli, respectively, farmers in Ghana only obtain a maximum of 7.3Mt/ha relative to the country's potential of 30mt/ha [36]. The yields of tomatoes in Burkina Faso and Niger are more than double the yields in Ghana.

Several authors and institutions have examined the possible causes of the low productivity of vegetable farmers [37–40]. Their findings include among others; biotic (arthropod pests, fungal, viral, bacterial) and abiotic (high temperature, erratic rainfall, poor soil, etc) stresses, and inadequacies in public and private investment in yield-enhancing technologies [41]. Nonetheless, literature argues that apart from these constraints, technical inefficiencies and production risks often associated with farming practices reduce vegetable productivity in developing countries. [42]. Efficiency refers to the relationship between maximum attainable and realisable yields subject to prevailing production technologies, input prices and optimum behaviour of the farmer. Efficiency can either be technical or allocative and a combination of the two is economic efficiency. [43] defined technical efficiency (TE) as the ratio of actual output to maximum output (or the ratio of minimum input to actual output).

Studies involving stochastic frontier analysis in technical efficiencies in agricultural production have increased in recent years owing to its prudence in identifying the determinants of farm output [31,44–48]. The stochastic production function by [49,50] is noted to be more realistic for estimating technical efficiency than the deterministic frontier originally pioneered by [51,52]. Nonetheless, one important aspect of agriculture production not adequately addressed by the stochastic frontier approach is production risk. On that basis, the extent of technical efficiency is often compromised in studies of farm performance in risky environments. Risk influences input decisions and production outputs. Just and Pope [44] led the way in understanding the stochastic function under risk conditions. Their research considers production risk to measure the output variance while focusing on a specification that permits inputs to be either risk-increasing or risk-decreasing [53].

To date, no comprehensive study has incorporated production risk in the stochastic frontier model to assess the technical efficiency of vegetable production in Ghana. Most technical efficiency studies rely on the conventional stochastic frontier technique to contribute to the food

policy discourse [54–58], which fails to account for production risk [59]. This study overcomes this shortcoming by incorporating production risk to assess the technical efficiency of dry season vegetable production in the Upper East Region of Ghana. The study employs the SFA with flexible risk properties to estimate the extent of technical inefficiencies attributed to production risk. Thus, dry season vegetable production variability is assessed from two angles—production risk and technical inefficiency. A single-stage maximum-likelihood estimation is used to estimate the mean output value, production risk and technical efficiency models.

The outcome of this study expects to achieve at least three key reasons. First, vegetable farming involves intensive production with a low fallow ratio, greater use of inputs such as labour and capital and higher per unit land due to shorter growing seasons and multi-harvest [60]. Moreover, frequent management of pests and diseases and careful attention to soil fertility and water management are needed in vegetable production compared to other crops [61]. Therefore, arriving at optimal production decisions can help farmers grow with higher TE to meet the existing domestic demand deficits. Second, the study's results will contribute to efficient vegetable production and support public policy initiatives. Finally, the study is crucial for the Ghanaian vegetable sector to maximize profit and achieve food security. The next section of the paper presents the literature review. The materials and methods follow in section three. This continues with the results and discussion in section four and finally, section five provides the conclusions and policy recommendations.

## 2. Literature review

Vegetables play an essential role in nutritious diets in the world [62,63], requisite for the fight against several diseases [64,65]. The rising demand for vegetable products is attributed to the consumers' preferences, heightened health awareness and increasing population [16]. On a per capita basis, global vegetable production has grown by about 60% in the last two decades compared to the decade earlier (1999–2000). However, per capita vegetable availability has consistently shrunk below the recommended consumption levels due to low productivity and population growth. In 2015, 81 countries, representing 55% of the global population did not achieve average per capita vegetable availability to meet the minimum target of 240 per capita per day [3,21,66,67]. Development policies and agencies seek to stimulate crop production to match consumption through various interventions and investments but these initiatives tend to prioritise the demand side over the supply side [3]. Most, policy and research efforts have typically not prioritised horticultural crop production [16,68,69].

Vegetables are sensitive to extreme weather conditions, such as high temperatures and low soil moisture, which are responsible for low yields [70,71]. The harsh weather conditions in the form of drought, rising temperature levels, erratic rainfall patterns, and floods limit water availability for vegetable production [71]. Higher temperatures adversely reduce soil moisture and help pests and diseases to multiply, thereby reducing vegetable growth and productivity [72]. Decreased precipitation and increased temperature lead to serious crop-water stress situations, reduced water available for irrigation and increased evapo-transpotation [73]. A study by [74] found that climate variability; especially temperature and rainfall variability are the cause of the reduction in tomato yield in Ghana. This was confined by a study by [75] who concluded that the impact of climate change on vegetable production is higher in Ghana than in Nigeria and Uganda.

Studies by [76,77] state that several possible pathways exist to decrease climatic risks, sustainably increase productivity and enhance resilience in vegetable farming systems. Irrigation is a climate-smart agriculture strategy for crop production resilience for achieving stability in food security, and employment opportunities, meeting social needs, and reducing poverty

[78,79]. Irrigation farming has a positive and significant effect on crop revenues for its pre-paredness and ability to use improved farm inputs, enabling farmers to shift from staple to cash crop production [80]. Factors such as socio-demographic (age, sex, marital status family size), socio-economic (educational level, land size holding, oxen holding and farmer experience), and institutional characteristics (credit access, extension service and input use of household head) positively affect the uses of irrigation technology.

In Ghana, dry-season irrigation production is critical in employment creation, generation of income and food provision [81]. Irrigated agriculture practices support more than 40% of global food production as land under irrigated agriculture is about two times more productive than rainfed agriculture [82]. However, resource use efficiency in irrigated systems is much lower than the achievable potential [83]. A study by [84] indicated that factors such as age, sex, education, extension visits, farmer association membership, timing of production, and access to marketing significantly influenced farmers' vegetable production in Nothern Ghana. Again, [42] stated that land, hired labour, family labour fertilizer, and pesticides affect the technical efficiency of vegetable production in Ghana.

Improving resource use efficiency (both technical and allocative) and increasing productivity are the right steps to enhance the growth of crops' output. Technical efficiency approach is commonly employed to identify the determinants of a single crop as in cabbage and potato production in South Africa [85,86], okra production in Nigeria [87], onion production in Pakistan [7], and tomato production in Ghana [88]. In all the technical efficiency studies the output (in kg) is often the dependent variable. Few studies have combined two or more products as a dependent variable in a technical efficiency study [55,89]. Technical efficiency studies on the products of the whole farm, rather than specific products usually apply monetary value (either income or revenue) as an output variable instead of physical quantities (in kilogram) [55,90–94]. For example, [55] obtained a mean technical efficiency score of 66.9% for vegetable farmers in Cameroon, implying that about 33% of potential vegetable output was lost due to only technical inefficiency.

However, studies that do not consider the technical inefficiencies arising from limitations in farming practices and the risk-averse behaviour of farmers result in inflated and biased estimates [59]. A study by [38] combined production risk and technical efficiency in a stochastic frontier to analyse bean production in Bangladesh. Their result reveals that the risk of bean production increases with farm labour, irrigation and farm size, while fertilizer reduces the risk of bean production. Also, [95] study of maize farmers in Ethiopia reveals that fertilizer and ox plough days reduce output risk while labour and improved seed increase output risk. Their study found a mean technical efficiency score of 48%, which meant that both production risk and technical inefficiencies of the farmers prevented the maize farmers from realizing their frontier output. Also, [39] obtains an average maize production score of 62% in Ghana. Factors such as seed and labour are negatively related to maize production while land and cost of intermediate inputs are risk-increasing inputs. Despite several studies that have integrated production risk into the stochastic frontier approach to examine technical efficiency of crops [38,39,95], there remains a research gap regarding production risk and technical efficiency, particularly in vegetable production in Ghana.

## 3. Material and methods

### 3.1 The study area

The focus of this study is dry season vegetable farmers in the Upper East Region of Ghana. The total area of the region is about 8,842 km$^2$, representing about 3.7% of the country's total land mass, inhabited by 1,046,545 people with a population density of 118 persons/km$^2$ [96]. Like

the entire northern Ghana, the Upper East Region experiences a unimodal rainfall season with annual rainfall ranges between 700 mm and 1010 mm, with peak rainfall in late August and ending in early October. The annual evapotranspiration is higher than precipitation, making water storage reservoirs a limited source for agricultural activities during dry seasons. The long spell of dry season from November to mid-June is accompanied by dry cold and dusty "harmattan" winds from November to February. The months of March to May are, however, characterized by dry hot temperatures, sometimes above 40˚c. The natural vegetation is the savannah woodland with scattered drought-resistant trees and grasses [56].

Like most parts of the country, smallholder agriculture with a per capita land size averaging less than 2 hectares dominates the people's livelihood activities. The main types of soils are sandy clays, clay loamy, and sandy loam. Crops grown in the rainy season include millet, guinea corn, maize, groundnut, beans, and sorghum. Rice is however, grown in both rainy and dry seasons, though on a small scale during the dry season along with vegetables like tomatoes, pepper, onions, garden eggs, cabbage, "aleafu" and "bito" leaves at the irrigation sites.

The prospects for dry season agriculture to reduce poverty and increase food security, are motivations for successive governments to establish dams and dugouts to boost all-year-round farming in northern Ghana. Two large irrigation dams, the Tono and Vea dams, as well as small dams, dugouts are spread throughout the region to support agriculture. Besides, the annual run-off passes through several natural water bodies such as rivers and streams in the region. The White Volta, in particular drains several water bodies in the northern part of the country, flowing southward. The White Volta River basin in particular connects with the Bagre dam in Burkina Faso [97]. Annually, managers of the Bagre dam in Burkina Faso spill excess water between September and October, usually causing floods and damaging livelihoods and putting lives at risk in northern Ghana with the Upper East Region largely affected [98]. The abundance of grassland supports livestock like cattle, goats, sheep and pigs under a semi-intensive system. Poultry such as fowls and guinea fowls are also reared on a small scale.

### 3.2 Population, sampling procedure and sample size

The study employed data from a comprehensive survey of dry-season irrigated vegetable farmers in the Upper East Region. The Upper East Region was chosen because vegetable farming constitutes the main livelihood activity of the people during the dry season.

A multi-stage sampling design was applied to sample dry-season vegetable irrigators for the study. In the first stage, seven (7) out of the fifteen (15) municipals and districts where vegetables are cultivated in the region were purposively sampled and zoned into three to constitute the Primary Sampling Unit (PSU). That is, the PSU comprises the Central zone (Bolga Municipal, Talensi and Bongo Districts), Eastern zone (Bawku East Municipal, Binduri and Bawku West Districts) and Western zone (Kasena Nankana East Municipal). All the sampled districts have varied sources of water bodies that farmers use for vegetable cultivation during the dry season. The western zone, though covered only by Kasena Nankana East Municipal, is an area with the largest reservoir (Tono dam) in the region which is managed by the Irrigation Company of Upper Region (ICOUR) and is used by many dry-season vegetable farmers in the region.

In the second stage, eight (8) communities per farming zone were randomly selected from each of the three (3) zones, making up 24 communities. A complete list of vegetable farmers from the sampled communities provided by the respective district office of the Ministry of Food and Agriculture (MOFA) constituted the Secondary Sampling Unit (SSU).

Then, Cochran's sample size formula [99] was employed to determine the required number of dry-season vegetable farmers for the study. The generalized Cochran formula is given as:

$$n_o = \frac{Z^2 pq}{e^2} \tag{1}$$

Where e (desired precision) = 0.05, at 95% confidence interval (z = 1.96), p = 0.5 (proportion of the population size), q = 1 − p. This gave a Cochran's sample size value ($n_0$) of 385. Now, adjusting with a total sample population of 1475 vegetable farmers in the formula in Eq (2) gives a sample size of 322 farmers, considering a non-response rate of 5%.

$$n = \frac{n_0}{1 + \frac{(n_0 - 1)}{N}} \tag{2}$$

With the support of one Agricultural Extension Agent (AEA) from each district, 322 farmers were systematically sampled from the 24 communities across the 7 districts based on a systematic random sampling method (Table 1). The systematic random sampling method involves selecting sample units at specific intervals. A similar sampling technique was

**Table 1. Sample size distribution of dry-season vegetable farmers.**

| Zone | District/Municipal | Community | Sample Size | % |
|---|---|---|---|---|
| Central | Bolga | Yikene | 19 | 5.90 |
| | | Zaare/yorogo | 15 | 4.66 |
| | | Sumbrungu/Dindubisi | 8 | 2.48 |
| | | Bolga-Nyariga | 18 | 5.59 |
| | Bongo | Bongo-Nyariga | 13 | 4.04 |
| | | Gowrie/Vea | 14 | 4.35 |
| | Talensi | Pwalugu | 10 | 3.11 |
| | | Baare/Yinduri | 9 | 2.80 |
| **Sub-Total** | | | **106** | **32.93** |
| Eastern | Bawku West | Kamega/Teshie | 9 | 2.80 |
| | | Timonde/Yarigu | 13 | 4.04 |
| | | Saka/Ampalug | 12 | 3.72 |
| | Binduri | Kumpalgoga | 10 | 3.11 |
| | | Binguri | 17 | 5.28 |
| | | Azum Sapelga | 30 | 9.32 |
| | Bawku East | Kpalwega | 15 | 4.66 |
| | | Modnori | 16 | 4.97 |
| **Sub-Total** | | | **122** | **37.90** |
| Western | Kasena Nankana East | Biu | 16 | 4.97 |
| | | Gaani | 8 | 2.48 |
| | | Korania | 12 | 3.72 |
| | | Yoobgannia | 15 | 4.66 |
| | | Bonia | 13 | 4.04 |
| | | Wuru | 10 | 3.11 |
| | | Yigbwannia | 9 | 2.80 |
| | | Chuchuliga | 11 | 3.42 |
| **Sub-Total** | | | **94** | **29.20** |
| **Grand Total** | | | **322** | **100** |

Source: MoFA and ICOUR, and Own survey (2020).

employed by the Ghana Statistical Service for the Ghana Living Standard Surveys [100] and adopted by [101] for their study in Ethiopia. Under the direct supervision of the researcher, primary data was collected from the dry-season vegetable farmers between May and November 2019.

Experienced enumerators across the various districts in the region were recruited and trained on the use of smartphone technology and deployed for the data collection under the supervision of the corresponding author of this paper and three coordinators (one from each zone) were recruited for the data collection. Data were collected via face-to-face personal interviews of vegetable farmers using tablets. The data collected was mainly quantitative and covered the socio-economic characteristics of the farmers as well as the inputs, outputs, prices and cost of production of vegetable production. The Open Data Kit (ODK) platform was used and data was uploaded daily.

### 3.3 Theoretical concepts of stochastic frontier approach

Mathematically, the production function for a fully technically efficient farmer (without inefficiency) in a production process is given as:

$$Y_i = f(x_i\beta) \tag{3}$$

However, due to challenges in agriculture production, no production process is 100 per cent efficient. Farmers are often exposed to risks and other shocks. As such, [50,102] proposed a stochastic production function to include the error term which was subsequently adopted by [103] given as:

$$Y_i = f(X_i; \beta) + \varepsilon_i \tag{4}$$

Where the subscript $i$ denotes the *ith* decision-making unit (DMU), $i = 1, 2, 3, \ldots n$, $Y_i$ is the output of the *ith* farmer, $x_i$ is a vector of $k \times 1$ input and other explanatory variables of the *ith* farmer, $\beta$ is the vector of fixed factor (technology) parameter, $f(.)$ is the best production frontier and $\varepsilon_i$ is the error term decomposed into two independent variables $u_i$ and $v_i$ where $\varepsilon_i \equiv v_i - u_i$.

The $v_i$ error component is symmetric and represents random error beyond the farmer's control (weather, disease, luck, measurement errors, omitted variables and other noises). The random component is assumed to be identically and independently normally distributed, with zero mean and constant variance $[v_i \sim N(0, \sigma_v^2)]$. The inefficiency component ($u_i$) on the other hand, is non-symmetric and assumed independently distributed with the random error ($v_i$) as well as meets the conditions of $u_i \geq 0$. It also measures the deviation of the observed technical efficiency ($Y_i$) from the technical efficiency of the "best" performing farm on the production frontier $[f(x; \beta) + v]$.

Generally, smallholder farmers in developing countries often fail to adopt or partially adopt new technologies due to some perceived risk profiles associated with such technologies. For instance, the high costs of inputs like fertilizer, agrochemicals, and labour make farmers take certain precautionary production decisions to minimize costs. Also, uncertainties associated with the marketing of perishables farm produce like vegetables also restrain some farmers from spending resources beyond certain thresholds on technologies even when these investments could provide higher returns on the land and labour than the pre-existing technologies on the grounds of risk [104]. In such instances, applying the conventional stochastic frontier model to assess technical efficiency does not capture these perceived risk behaviours of the farmers [105]. Hence, ignoring these marginal effects of the risks associated with the behaviour of the farmers' decisions may result in erroneous estimates [102]. [36] study of fish farms in

Nigeria found a mean technical efficiency score of 0.79 with the inclusion of flexible risk and 0.92 without the use of production risk function. Most studies dealing with production risks employ the formula of [44]. The inclusion of the production risk function helps to cater for the risk behaviour of farmers in decision-making. The production risk function is given as:

$$Y_i = f(x; \beta) + \varepsilon = f(x; \beta) + g(z; \theta)v \tag{5}$$

[53] made two further modifications to the formula. First, the inefficiency function was added to the Pope and Just's model. Second, the inefficiency function was multiplied by Pope and Just's model, to make it consistent with [50,102]. This study employs the additive heteroskedastic error model structure of [53,103], for determining the technical efficiency score of dry season vegetable production in the Upper East Region of Ghana. The model conforms to the [44] model specified:

$$Y_i = f(x; \beta) + g(z; \theta)v - q(w; \delta)u. \tag{6}$$

Where: $f(x; \beta)$ is the mean output function, $g(z; \theta)v$ is the production risk function and $q(w; \delta)u$ is an inefficiency function. The inefficiency function captures the relationship between the technical efficiency of the vegetable farmers and different demographic, socio-economic, farm attributes, institutional and managerial characteristics of the farmers, where the $w$ is the vector of *ineficiency* factors and $\delta$ is the parameter to be estimated. Eq (6) is synonymous with the general conventional SFA model, which exhibits heteroskedasticity where $\delta_v^2 = \exp(z; \theta)$ and $\delta_u^2 = \exp(w; \delta)$ represent $v$ and $u$, respectively [56].

From the estimates of the inputs, the extent of the inefficiency effects, the mean output of the *ith* farmer is given by:

$$E\left(\frac{Y_i}{x, u}\right) = f(x; \beta) - g(w; \theta)u \tag{7}$$

The inefficient dry-season vegetable farmers are those with positive deviations, which lie below the frontier while the efficient farmers are those with zero deviations which lie on the production frontier [106]. The technical efficiency (*TE*) therefore, is the ratio of the observed technical efficiency of a given vegetable farmer to the technical efficiency of a fully efficient farmer with the same input vectors [107] stated as:

$$TE_i = \frac{E(Y_i/x, u)}{E(Y_i)/x, u - 0} = \frac{f(x; \beta) - g(z; \theta)u}{f(x; \beta)} \tag{8}$$

Hence, $TE_i = 1 - \frac{g(z;\theta)u}{f(x;\beta)}$, but since technical inefficiency $(TI_i) = \frac{g(z;\theta)u}{f(x;\beta)}$, then

$$TE_i = 1 - TI_i \tag{9}$$

Since $E(TE_i) = f(x; \beta)$ and $V(y) = g^2 (z; \theta)$, then, the marginal technical efficiency (production risk) is a partial derivative of the risk function with respect to the inputs $(x_i)$ given as;

$$\frac{\partial Var(\pi)}{\partial x_i} = 2g(z; \theta)\frac{\partial g}{\partial x_i} \tag{10}$$

In that case, where $2g(z; \theta)\frac{\partial g}{\partial x_i} > 0$, such an input is risk increasing, if $2g(z; \theta)\frac{\partial g}{\partial x_i} < 0$, the input is risk decreasing and if $2g(z; \theta)\frac{\partial g}{\partial x_i} = 0$, such an input is risk neutral. The marginal production risk, therefore has a directional sign that can be positive, negative or zero to represent

risk increasing, risk decreasing, and risk neutral, respectively. Therefore, a risk-averse farmer will allocate more resources to less risky income sources than a risk-neutral farmer [53].

To ascertain the functional form which is appropriate for the data, a log-likelihood ratio test was performed on the null hypothesis that the Cobb-Douglass functional form is appropriate for the data against the alternative hypothesis that the trans-log functional best fit for the data given as:

$$\lambda = 2(lnL_{Cobb-Douglass\ functional\ form} - lnL_{Translog\ functional\ form}) \qquad (11)$$

Where; $\lambda$ is distributed as a Chi-square with a degree of freedom equal to the number of independent variables including the constant. The test results reject the Cobb-Douglass functional form in favour of the trans-log if the test statistic ($\lambda$) is greater than the appropriate Chi-square critical value. In other words, a hypothesis test was conducted on whether $\beta_{jk} = 0$. We found that the trans-log functional form best suits the stochastic frontier for the mean function of dry season vegetable farmers in the Upper East region of Ghana stated as:

$$lnY_i = \beta_0 + \sum_{j=1}^{5} \beta_j lnx_{ji} + 0.5 \sum_{j=i}^{5} \sum_{k=i}^{5} \beta_{jk} lnx_{ji} lnx_{ki} + v_i - u_i \qquad (12)$$

Where $Y_i$ is defined as the value of vegetables (GHS) produced by the farmer in the $ith$ area, which is log-transformed for $i = 1,2,..322$, $x_{ji}$ denotes the vector of cost (GHS) of different production inputs $j$ used by the $ith$ vegetable farmer. All inputs and outputs as shown in Table 2 were evaluated on a per per-hectare basis, where $x_1$ represents the cost of labour (GHS), $x_2$ denotes the cost of seed (GHS), $x_3$ denotes the cost of fertilizer (GHS), $x_4$ denotes the cost of agrochemicals, $x_5$ denotes the average cost of irrigating the farm (water levy, running and maintenance cost of pump or cost of digging wells and machinery depreciation). The cost of inputs is estimated separately for each vegetable cultivated on a farm and summed for all crops for the farmer. The explanatory variables identified are among the most critical hampering smallholder irrigated vegetable cultivation in Sub-Saharan Africa including Ghana [55,94,107].

This study adopted the measurement of labour in wages per day contrary to man-days employed by [108], who measured labour according to sex and age group of labourers. Wages per day give a more accurate measurement of the cost of labour than man-days since some activities may take a few hours (less than a day) making it cumbersome to aggregate. Typical of smallholder farmers in SSA, dry season vegetable farmers in the study cultivate more than one type of vegetable in the season. The value of each vegetable was obtained by multiplying the quantity of the vegetable by the respective unit price and summing up to arrive at the total output value. Studies such as [59], and [93] employed similar methods to estimate technical

**Table 2. Description of variables in the mean technical efficiency function.**

| Variable | Measurement | Symbol | Description | Expected sign |
|---|---|---|---|---|
| Output Value | GHS | $Y_1$ | Output Value per ha | |
| Labour cost | GHS | $\beta_1$ | labour per ha | + |
| Seed cost | GHS | $\beta_2$ | seed per ha | + |
| Fertilizer cost | GHS | $\beta_3$ | fertilizer per ha | + |
| Agrochem. cost | GHS | $\beta_4$ | agrochem. per ha | + |
| Irrigation cost | GHS | $\beta_5$ | irrigations per ha | + |

Authors construct, (2020).

efficiency of vegetable production. Technical efficiency studies on the products of the whole farm, rather than specific products usually apply the monetary value of output instead of the physical quantities due to differences in per unit's prices [56,90,91,94,108]. A similar analysis was employed in this study by aggregating the value of outputs of all the vegetables produced by the farmer.

## 3.4 Estimation of elasticities

With the trans-log functional form, the elasticities for the cost of the various inputs were normalized with land (ha) and scaled by the respective sample means according to [57] given as:

$$\frac{\partial LnE(Y_i)}{\partial Lnx_{ji}} = \left\{ \beta_j + \beta_{ji}Lnx_{ji} + \sum_{k \neq j} \beta_{jk}Lnx_{ki} \right\} \tag{13}$$

In such a situation, the first-order coefficients were expressed as the elasticities of the technical efficiency with respect to the corresponding input cost specification. The sum of the coefficient of the input variables is the scale elasticities ($\rho$), which measures the return to scale for the dry season vegetable industry in the Upper East region. It determines the percentage change in technical efficiency as a result of a 1% change in all costs of input variables. The return to scale is increasing return to scale (IRS) if ($\rho > 1$), decreasing return to scale (DRS) if ($\rho < 1$), or constant return to scale (CRS) if ($\rho = 1$). Table 2 illustrates a description of the variables in the mean technical efficiency function.

## 3.5 Marginal production risk model

To be able to appropriately determine the effect of the decision of the farmer on the choice of inputs on the mean output and output variance, the random error term $\varepsilon = v - u$ was parameterized to include production risk $g(z_i; \varphi) v_i$ and inefficiency $q(w; \delta)u$ such that:

$$\varepsilon = g(z_i; \varphi)v_i - q(w_i; \delta)u_i \tag{14}$$

The production risk function was linearized as:

$$g(z_i; \varphi)v_i = \varphi_0 + \sum_{m=i}^{5} \varphi_m x_{mi} \tag{15}$$

Where $\varphi_0$ represents the intercept, $x_m$ denotes the input variables, and $\varphi_m$ signifies the risk parameters to be determined, where if the estimate is positive, such input is risk increasing, negative estimate indicates a risk decreasing and zero means risk-neutral input or such input does not have any effect on the production risk of dry season vegetable farms in the study area [38,44]. The study expects all inputs in the model to affect the output value of vegetables as some inputs may reduce the level of output value of vegetables whereas others may increase it as shown in Table 3.

## 3.6 Specification of inefficiency model

Given the functional specifications in Eq (16), the effects of the inefficiency variables($u$) hypothesized in literature by [103] to explain the variation in technical inefficiency given as:

$$q(w_n; \delta)u = \delta_0 + \sum_{j=1}^{19} w_{ni}\delta_n + \alpha_i D_{ki} \tag{16}$$

Where $\delta_0$ denotes the intercept, $\delta_n$'s ($n = 1, 2, 3, \ldots, 19$) are the unknown inefficiency

**Table 3. Description of variables used in the production risk function.**

| Variable | Measurement | Symbol | Description | Expected sign |
|---|---|---|---|---|
| Labour cost | GHS | $\psi_1$ | Labour per ha | - |
| Seed cost | GHS | $\psi_2$ | Seed per ha | - |
| Fertiliser cost | GHS | $\psi_3$ | Fertilizer per ha | + |
| Agrochem. cost | GHS | $\psi_4$ | Agrochem. per ha | - |
| Irrigation cost | GHS | $\psi_5$ | Irrigations per ha | + |

Authors construct, (2020).

parameters to be estimated. The *w's* are the specific continuous variables while the *D's* are dummy variables that explain the inefficiencies. These variables are defined and measured with their a priori expected signs in Table 4. It must be stated that model 16 only becomes relevant if the inefficiency effect is stochastic and follows particular distributional properties [109]. A negative sign of the variable suggests a positive effect on technical inefficiency (TE) and vice versa [71].

## 3.7 Statement of hypothesis

The null hypothesis ($H_o$) of no influence against alternative ($H_A$) of the significant influence of explanatory variables on the technical efficiency of dry-season vegetable farmers is stated as:

**Table 4. Description of inefficiency variables that influence vegetable production.**

| Variable | Measurement | | Symbol | Description | Sign |
|---|---|---|---|---|---|
| Irrigations | Number | | $w_1$ | Frequency of irrigating farm | + |
| Household size | Number | | $w_2$ | Household members | + |
| Experience | Years | | $w_3$ | Years of farming | + |
| Farmer's age | Years | | $w_4$ | Age of farmer | + |
| Age squared | Years | | $w_5$ | Age squared | - |
| Extension visits | Number | | $w_6$ | Number of visits | + |
| Gender | Yes = 1 if male | No = 0, otherwise | $w_7$ | Gender of farmer | + |
| Credit access | Yes = 1 | No = 0 | $w_8$ | Farmer received credit | + |
| Irrigation Tech | | | | | |
| Manual | Yes = 1 | No = 0 | $w_9$ | Manual irrigation of farm | + |
| Gravity fed | Yes = 1 | No = 0 | $w_{10}$ | Gravity-fed irrigation of farm | - |
| Water pump | Yes = 1 | No = 0 | $w_{11}$ | Pump irrigation of farm | - |
| Farmer's Edu. | | | | | |
| No formal edu. | Yes = 1 | No = 0 | $w_{12}$ | Not attained formal education | - |
| Basic edu. | Yes = 1 | No = 0 | $w_{13}$ | Attained basic level | + |
| SHS/Tech/Voc. | Yes = 1 | No = 0 | $w_{14}$ | Attained SHS/Tech/Voc. | + |
| Tertiary edu. | Yes = 1 | No = 0 | $w_{15}$ | Attained tertiary level | + |
| Crop Type | | | | | |
| Pepper | Yes = 1 | No = 0 | $w_{16}$ | Cultivated pepper | - |
| Onions | Yes = 1 | No = 0 | $w_{17}$ | Cultivated onions | - |
| Tomatoes | Yes = 1 | No = 0 | $w_{18}$ | Cultivated tomatoes | - |
| Garden eggs | Yes = 1 | No = 0 | $w_{19}$ | Cultivated garden eggs | - |

Source: Authors construct, (2020).

$H_o$: $\beta_1 = \beta_2 = \beta_3 + \ldots = \beta_{19} = 0$; farmer's age, age squared, extension visits, gender, education (no education, basic, SHS/Tech/Voc, Tertiary), household size, irrigation technology (manual, water pump, gravity-fed), frequency of irrigations, credit access, experience and type of vegetable crop do not influence the technical efficiency of dry-season vegetable farmers in the Upper East region of Ghana.

$H_A$: $\beta_1 > \beta_2 > \beta_3 + \ldots > \beta_{19} > 0$; farmer's age, age squared, gender, education (no education, basic, SHS/Tech/Voc, Tertiary), extension visits, household size, irrigation technology (manual, water pump, gravity-fed), frequency of irrigations, credit access, experience and type of vegetable crop have a significant positive influence on the technical efficiency of dry-season vegetable farmers in the Upper East region of Ghana.

## 3.8 Hypotheses test of model fitness

The following hypotheses were tested using the generalized likelihood ratio test to ascertain the best fit of the models specified; the relevance or otherwise of the risk component; the presence of technical inefficiency; and the importance or otherwise of each of the variables identified in the model.

The test for the functional form was given as $H_{0:}$ $\beta_{ij} = 0$, the null hypothesis that the coefficient of the second order in the model is zero or otherwise. The test statistic determines which functional form is appropriate to adopt for the study, that is either the trans-log or the Cobb-Douglas functional form.

$H_0$: $\theta_1 = \theta_2 = \ldots \theta_5 = 0$. The null hypothesis states production risk relating to input factors does not explain the variability of vegetable output value.

$H_0$: $\lambda = 0$. The null hypothesis states that the inefficiency effect is absent from the model at every level. That is, the variance of the technical inefficiency term is absent, meaning the exogenous factors should be incorporated into the mean output function and estimated as traditional average normal responses using Ordinary Least squares (OLS). However, if $\lambda > 0$, it means the inefficiency effects are present in the model, as such, the stochastic frontier model should be applied.

$H_0$: $\delta_1 = \delta_2 = \ldots \delta_{19} = 0$. The null hypothesis specifies that farm-specific exogenous factors do not jointly influence technical efficiency.

$H_0$: $\Sigma\beta_i = 1$. The hypothesis is that the sum of the coefficients of the input variables is a constant return to scale (CRS).

## 3.9 Hypotheses validation and decision criteria

The above hypotheses were tested using the generalized likelihood ratio statistic stated as:

$$\text{LR} = n[\ln(\text{L}(\text{H}_0) - \ln(\text{H}_1)] \tag{17}$$

Where: $L(H_0)$ and $L(H_1)$ represent the null and alternative hypotheses, respectively. LR has a chi-square distribution if the given null hypothesis is true with a degree of freedom equal to the number of parameters assumed to be equal to zero in the null hypothesis. The third hypothesis was examined with a mixed chi-square [110]. The fifth hypothesis is tested with probability (p-value). The effect of explanatory variables on technical inefficiencies of dry-season vegetable farmers was also tested with probability (P-value). STATA 15.1 software was employed for the data analysis.

### 3.10 Ethics approval statement

This paper is an excerpt from my PhD thesis in the Department of Agricultural Economics and Agribusiness, University of Ghana. The study was approved by the Ethics Committee for Basic and Applied Science (ECBAS) of the University of Ghana, permit number ECBAS 0012/ 18–19.

## 4. Results and discussion

### 4.1 Summary statistics of the explanatory variables

Dry-season vegetable production is attractive to farmers because of its high demand and returns on investment, which supports farm families to improve their income and food security level. Table 5 describes summary statistics of inputs and output cost and socioeconomic

**Table 5. Summary of selected variables in frontier and inefficiency models.**

| Continuous variables | | | | | |
|---|---|---|---|---|---|
| **Frontier var.** | **Measurement** | **Mean** | **Std.** | **Min.** | **Max.** |
| Veg. value | GHS/ha | 8449.86 | 8172.00 | 105.00 | 39100.00 |
| Labour cost | GHS/ha | 4182.00 | 2951.00 | 321.00 | 19958.00 |
| Seed cost | GHS/ha | 245.00 | 248.00 | 10.00 | 2400.00 |
| Fertilizer cost | GHS/ha | 359.00 | 292.00 | 10.00 | 2120.00 |
| Agrochem. Cost | GHS/ha | 271.00 | 267.00 | 1.00 | 1600.00 |
| Irrigation cost | GHS/ha | 807.00 | 807.00 | 40.00 | 6120.00 |
| **Inefficiency var.** | | | | | |
| Irrigations | Watering no. | 51 | 42 | 10 | 255 |
| Household size | Number | 7 | 3 | 1 | 35 |
| Experience | Years | 9 | 7 | 1 | 45 |
| Farmer's age | Years | 41 | 11 | 16 | 86 |
| Age squared | Years | 1872 | 1006 | 256 | 7396 |
| Extension visits | Number | 2 | 2 | 0 | 7 |
| **Dummy var.** | **Measurement** | | | **% of farmers with 1** | **% of farmers with 0** |
| **Irrigation tech** | | | | | |
| Manual | Yes = 1 | No = 0 | | 16.80 | 83.20 |
| Gravity fed | Yes = 1 | No = 0 | | 18.60 | 81.60 |
| Water pump | Yes = 1 | No = 0 | | 64.60 | 35.40 |
| Gender | Yes = 1 | No = 0 | | 90.30 | 9.70 |
| Credit access | Yes = 1 | No = 0 | | 18.00 | 82.00 |
| **Farmer's edu.** | | | | | |
| No formal edu. | Yes = 1 | No = 0 | | 37.60 | 62.40 |
| Primary/JHS | Yes = 1 | No = 0 | | 41.30 | 58.30 |
| SHS/Tech/Voc. | Yes = 1 | No = 0 | | 14.00 | 86.00 |
| Tertiary | Yes = 1 | No = 0 | | 7.10 | 92.90 |
| **Crop type** | | | | | |
| Pepper | Yes = 1 | No = 0 | | 68.00 | 32.00 |
| Onions | Yes = 1 | No = 0 | | 25.78 | 74.22 |
| Tomatoes | Yes = 1 | No = 0 | | 13.66 | 86.34 |
| Garden eggs | Yes = 1 | No = 0 | | 8.07 | 91.93 |

Source: Survey results, (2020).

variables used in the stochastic frontier model. The highest cost of production came from labour and irrigation (measured in GHS). The mean labour and irrigation costs were GHS 4,182.00/ha and GHS 807.00/ha, respectively. Other mean costs in the analysis included were seed (GHS 245.00), fertilizer (GHS 359.00), and agrochemical (GHS 271.00). The study did not measure the profitability of vegetable farming. Instead, the study determines the production risk and technical efficiency of the farmers. The mean total output value (dependent variable) was GHS 8,449.86/ha. The socioeconomic variables included in the inefficiency model were the irrigation frequency (or frequency of water applications) to the farm per month, irrigation technology, household size, age, extension visit and experience, education and vegetable type.

Ideally, the estimations should be based on the quantity (volume) of water, unfortunately, data on the quantity of water applied on the farm per month was not available. The study, therefore, relied on the frequencies of water application per month. The results show that a farmer requires between 10–40 applications of water per month. This depends on the type of soil structure, the growing stage of the crop and the type of irrigation technology the farmer adopts. The mean number of water applications per growing season (3 months) is 51. For instance, farmers who use motorized pumps and or gravity-fed apply water once every three intervals, while those who use the manual system (using watering cans, and calabashes buckets) apply twice (morning and evening) daily. The differences in technology account for the differences in the number of applications of water on the farm.

The volume of water applied using a pump or flood technology is more than a manual system. With the manual system, a small volume of water is applied to the crops per time, hence it has to be more frequent to avoid the effect of drought. The high temperature (sometimes above 40˚c) usually experienced during the day encourages high evapotranspiration and water stress, as such water is required to replenish as often as possible to maintain the required moisture level for the crops to flourish. A study by [111] recommends irrigation water between 4.1–5.6 mm per day or 0.3–0.4 per plant per day for tomato crops in the tropics. Farmers who use motorized pump systems of water application spend the bulk of their resources on the operation cost (fuel cost) and maintenance (repairs costs) of machinery. Farmers from the Bawku enclave (eastern zone) rely on temporal wells as their source of irrigation. These farmers spend their resources to dig temporary wells and use pumps to irrigate their farms [35]. Age is measured in years. The mean age of vegetable farmers is 42 years. Experience measures the number of years a farmer is involved in vegetable farming. The average experience of a farmer was 9 years. Extension visit was measured as the number of times an extension officer contacts a vegetable farmer when confronted with a challenge. The mean extension visits was 2 times. Irrigation technology, level of education and type of crop cultivated were all dummies.

## 4.2 Hypotheses test results

Table 6 presents the results of the maximum likelihood ratio tests of relevant hypotheses in the stochastic frontier analysis. The first test was the null hypothesis that the coefficient of the translog model's second–order ($\beta ij$) variables was zero. In other words, the test that the Cobb-Douglas model is appropriate for the data was rejected in favour of the translog model because the maximum likelihood ratio test statistic was greater than the tabulated (critical) value. The trans-log model was, therefore chosen as the best model for the data set thus, was employed to derive the estimates.

The second hypothesis test that production risk in the input choices does not exist in the production process was also rejected since the calculated statistic value was greater than the

**Table 6. Hypothesis test for model specification and statistical assumptions.**

| Null hypothesis | Test stat ($\lambda$) | Critical value | Decision rule | Conclusion |
|---|---|---|---|---|
| $H_{0:}\ \beta_{ij} = 0$ | 52.03 | 30.58[a] | Rejected | Trans-log is appropriate |
| $H_0$: $\theta_1 = \theta_2 = \ldots \theta_5 = 0.$ | 123.14 | 15.09[a] | Rejected | Production risk explains output variability |
| $H_0$: $\lambda = 0.$ | 169.48 | 9.50[b] | Rejected | Presence of Inefficiencies |
| $H_0$: $\delta_1 = \delta_2 = \ldots \delta_{20} = 0$ | 98.74 | 34.81[a] | Rejected | Exogenous variables affect variation. |
| $Ho$: $\sum Bi = 1$; <br> $Hi$: $\sum Bi \neq 1$ | | 0.00[c] | $Ho = 0$ rejected | Not CRS |

Source: Model result, (2020). $x^2\ 0.001^a\ mixed0.001^b,\ p-value^c$ Chi-square critical values are sig. at 1%.

critical Chi-square value. This implies that production risk associated with input choice is present in the data.

The third hypothesis that the inefficiency effect is not present, as such, the inefficiency effects are not stochastic was also rejected since the likelihood test statistic was greater than the critical value from the mixed square distribution table [110]. This means that employing the traditional average response function was inappropriate in representing the data. Rather, the data favours the stochastic frontier and inefficiency models. The fourth null hypothesis which states that all coefficients of the exogenous variables in the model explaining the inefficiency effects are zero was also rejected. This implies that the combined effect of the technical inefficiency variables are important determinant explaining the output variability of dry season vegetable production in the region. This does not mean, individually, the variables may not be significant.

The fifth hypothesis which states that the sum of the coefficients of the elasticities is equal to a constant was rejected. This implies that the sum of the coefficients of elasticities could either be in the range of Decreasing Return to Scale (DRS) or Increasing Return to Scale (IRS).

## 4.3 Maximum likelihood estimates of technical efficiency of dry season farmers

The maximum-likelihood estimates for the stochastic translog model are indicated in Table 7 below. The results show the mean output value function (upper rows), production risk function (middle rows), the variance parameter, the multicollinearity as well as heteroskedasticity test results (lower rows).

The Diagnostic Statistic of model fitness revealed a lambda ($\lambda = \sigma u / \sigma v$) of 1.904 in the function which is highly greater than zero, implying that the model and its distributional assumptions were chosen appropriately. The positive lambda equally indicates that the variations in the observed output values from the frontier are due to technical inefficiency and stochastic noise components, as such, the data cannot be estimated as a least square model. The results however, reveal that the deviation from the frontier attributed to technical inefficiency ($\sigma u$ = 0.456) is relatively higher than the deviations due to the idiosyncratic disturbance effects ($\sigma v$ = 0.239), attributed to measurement errors, adverse weather conditions, diseases and pest infestations, etc. The gamma ($\gamma$) value of 0.78 implies that 78% of the total variations in the output value of vegetables are attributed to technical inefficiencies in the area.

The Wald chi-square statistic (4871.390) is highly significant (1%), indicating the presence of a joint significance of the stochastic frontier model. The small value of the Variance Inflation Factor (VIF) of 1.630 reveals that the model is without multicollinearity [112]. Lastly, the

**Table 7. Maximum likelihood estimates of trans-log mean stochastic function.**

| Variable | Par. | Coeff. | Standard error |
|---|---|---|---|
| Constant | $\beta_0$ | 0.184*** | 0.042 |
| Lnlabour cost | $\beta 1$ | 0.434*** | 0.042 |
| Lnseed cost | $\beta 2$ | 0.099*** | 0.038 |
| Lnfertilizer cost | $\beta 3$ | 0.383*** | 0.048 |
| Lnagrochemical cost | $\beta 4$ | 0.321*** | 0.031 |
| Lnlrrigation cost | $\beta 5$ | 0.105*** | 0.032 |
| $0.5Ln(labour\ cost)^2$ | $\beta 6$ | 0.195 | 0.120 |
| $0.5Ln(seed\ cost)^2$ | $\beta 7$ | -0.156 | 0.102 |
| $0.5Ln(fertilizer\ cost)^2$ | $\beta 8$ | -0.086 | 0.095 |
| $0.5Ln(agrochemical\ cost)^2$ | $\beta 9$ | 0.055 | 0.035 |
| $0.5Ln(irrigation\ cost)^2$ | $\beta 10$ | 0.383*** | 0.076 |
| Lnlabour cost * Lnseed cost | $\beta 11$ | -0.114 | 0.074 |
| Lnlabour cost* Lnfertilizer cost | $\beta 12$ | 0.009 | 0.094 |
| Lnlabour cost* Lnagrochemical cost | $\beta 13$ | -0.040 | 0.052 |
| Lnlabour cost* Lnirrigation cost | $\beta 14$ | -0.238*** | 0.068 |
| Lnseedcost * Lnfertilizer cost | $\beta 15$ | 0.275*** | 0.068 |
| Lnseed cost* Lnagrochemical cost | $\beta 16$ | 0.046 | 0.053 |
| Lnseed cost* Lnirrigation cost | $\beta 17$ | -0.140** | 0.063 |
| Lnfertilizer cost* agrochemical cost | $\beta 18$ | 0.052 | 0.056 |
| Lnfertilizer cost* Lnlrrigation cost | $\beta 19$ | -0.071 | 0.061 |
| Lnagrochemical cost* Lnlrrigation cost | $\beta 20$ | -0.035 | 0.041 |
| **Production Risk variables** | **Par.** | **Coefficient** | **Standard error** |
| Lnlabour cost | $\psi_1$ | -0.001*** | 0.000 |
| Lnseed cost | $\psi_2$ | -0.000 | 0.001 |
| Lnfertilizer cost | $\psi_3$ | 0.002** | 0.001 |
| Lnagrochemical | $\psi_4$ | -0.003** | 0.001 |
| Lnlrrigation cost | $\psi_5$ | 0.000 | 0.000 |
| **Model diagnostics** | | | |
| $\sigma u$ | | 0.456 | |
| $\sigma v$ | | 0.239 | |
| Lambda ($\lambda = \sigma u/\sigma v$) | | 1.908 | |
| $\sigma^2 = \sigma_u^2 + \sigma_v^2$ | | 0.265 | |
| Gamma ($\gamma$) $= \sigma_u^2/\sigma_u^2 + \sigma_v^2$ | | 0.784 | |
| Wald chi2 | | 4871.390*** | |
| Mean VIF (multicollinearity) | | 1.630 | |
| Breusch-Pagan test (heteroskedasticity) | | 0.944 | |
| Log-likelihood function | | -90.165 | |
| Mean technical efficiency | | 0.730 | |
| Maximum TE | | 0.990 | |
| Minimum TE | | 0.070 | |
| Number of observations | | 322 | |

Source: Model results, (2020).

Note:

*** p<0.01,

** p<0.05,

* p<0.1.

outcome of the Breusch Pagan (BP) statistic test of 0.944 not being significant reveals the absence of heteroskedasticity.

The results of the maximum likelihood estimates show that the signs of all the variables included in the parameter estimates of the stochastic frontier translog model are according to the theory. Note that the various input costs were normalized with land (ha) and scaled by their respective sample means [103]. Therefore, the first-order coefficients are considered as elasticities of the output value of the vegetables with respect to the cost of the different inputs. In that case, the coefficient of each input cost is interpreted as elasticity [50]. The coefficient of labour, seed, fertilizer, agrochemical, and irrigation costs are positively related to the output value of vegetables and highly significant ($p < 0.01$), indicating that all the inputs have a reasserting influence on vegetable productivity in the region. This implies that a 1% increase in the investment cost on labour, fertilizer, agrochemicals, irrigation and seed increases the output value of vegetables per hectare by 0.434%, 0.383%, 0.321%, 0.105% and 0.099%, respectively.

The results further reveal that labour has the most important (with the highest elasticity) effect on the output value of vegetable production. This implies that investing more in labour yields higher returns. This is because vegetable production is labour-intensive [113]. Indeed, farm activities such as weeding, applying the right quantity of water, early detection of pests and controlling them, harvesting etc are necessary to maintain the quality of vegetable farms for effective fruiting for higher income [113]. The finding is consistent with the study by [90] of vegetable farms in Samsun Province in Turkey and [91] on vegetable-poultry integration in rural Tanzania.

Statistically, the coefficient of fertilizer is significant and positive, implying that using more fertilizer significantly increases productivity. Fertilizer is an important yield-boosting input. This was expected because fertilizer plays a major role in enhancing the yield of vegetables without which the crops may fail to fruit. Fertilizer replenishes nutrients in the soil, improves the chemical composition, enhances the activities of microorganisms, and improves the texture and structure of the soil for plant growth and development. This result is consistent with [39] study on maize productivity in Ghana. Vegetable farmers must constantly apply fertilizer on their farms to achieve the desired output.

Again, the coefficient of agrochemicals is statistically significant and positive implying using more pesticides significantly increases productivity. Appropriate application of agrochemicals like pesticides is key to vegetable production due to farm pest infestations. This finding is similar to [114] results on irrigated onion production in Ethiopia and on vegetable production in Ghana [42] but contrary to tomato production also in Ghana [88]. The study reveals that farmers in the study area who fail to invest in pesticides do not harvest high-quality vegetables and therefore suffer low income.

Further, irrigation cost shows a statistically significant and positive effect on productivity which indicates that the production of vegetables significantly increases with irrigation. Irrigation increases agricultural productivity and therefore improving irrigation technology must account for crop water requirements and conservation of soil fertility to boost productivity. This is consistent with [38] findings on bean productivity in Bangladesh, and [115] onion productivity in Ethiopia. Farming in the dry season would not be possible without irrigation infrastructure and equipment. Farmers with consistent irrigation water throughout the season get high yields thereby enhancing their farm income.

Lastly, the coefficient of seed is significant and positive implying using more improved seeds significantly increases vegetable productivity. The results agree with the findings of several studies including [56] on vegetable production from Ghana [95], maize production from Ethiopia, [55], and vegetable production from Cameroon but contradict [38] findings on bean

production in Bangladesh. Seed is expected to be positive because improved seeds give high yields, resulting in more income for the farmer.

The results of the marginal production risk estimates show that the signs of the coefficients of the production risk function all met the *a priori* expectation and are in line with the theory. The findings reveal information on the input factors causing output variability due to production risk decisions that are taken prior to production. While some inputs may be risk-minimizing, others may be risk-maximizing, thus providing an important guide to stabilize vegetable output in the study area.

The study shows that all variables in the production risk function influence vegetable production significantly. The estimates show that labour cost, seed cost and agrochemical costs decrease the variability of the output value of vegetables. That, seed cost is risk decreasing supports the a priori expectation that commercial seed results in less variation in the quantity and quality of vegetables produced, similar to the findings of [116] on subsistence farmers in the Kilimanjaro region of Tanzania.

Fertilizer cost also has a risk-increasing effect on vegetable production. This means using more fertilizer exposes the farmer to a more risky situation. The possible interpretation for this is that using more fertilizer reduces the soil's natural replenishing capacity, changes the soil's chemical behaviour, and makes it less fertile. Excessive fertilization changes the natural productivity of the soil by forming high salt condensation that harms microorganisms in the soil. Also, excessive fertilizer causes sudden growth of the plant and inadequate root system, tightening and acidification of soil, and leaching away soil nutrients [38]. The adoption of best management techniques along with optimal agricultural techniques increases productivity as asserted by [117].

The findings reveal that irrigation cost is a risk-increasing input. Therefore, investing in irrigation increases the risk of vegetable production. This can be explained by the fact that excessive usage of water damages vegetable plants by increasing the volume of salt on the surface soil thereby increasing the output variability of vegetables in the study location. Again, excessive irrigation limits the number of air pockets, causing a limited oxygen supply and inability of plants to breathe, and causing the water-logged roots not to grow properly [38]. The traditional irrigation technology can also be extremely labour intensive and can lead to waste of water and may require the establishment of efficient irrigation infrastructure which may be costly to implement.

The findings imply that a risk-averse vegetable farmer who intends to reduce the cost of production will most likely opt to increase the usage of more labour and apply more seed and agrochemicals, which may, in turn, affect the efficiency of vegetable production. However, that same farmer who may want to reduce losses due to production risk would be expected to use less fertilizer as well as reduce costs of investments in irrigation, which might have negative implications on vegetable output and eventually reduce the income for such a farmer. This finding lends credence to the fact that vegetable cultivation is a labour-intensive industry [112] and, as such plays a significant role in farm maintenance. It is important to stress that, a vegetable like pepper is a perennial, which requires constant weeding, application of fertilizer, agrochemicals, and irrigations for multiple fruiting, as such, requires more investments of capital and labour resources.

## 4.4 Elasticities of inputs

Table 8 indicates the elasticities of various inputs employed by the farmers for vegetable production in the study area. Total elasticity measures the Return to Scale (RTS) which is obtained by adding all the input elasticities of the production. Hence, the sum of elasticities equalled

**Table 8. Elasticity and return to scale.**

| Variable | Parameter | Elasticity |
|---|---|---|
| lSNlabour cost | $B_1$ | 0.434*** |
| lSNseed cost | $B_2$ | 0.099*** |
| lSNfertilizer cost | $B_3$ | 0.384*** |
| lSNagrochemical cost | $B_4$ | 0.321*** |
| lSNirrigation cost | $B_5$ | 0.105*** |
| Return to scale (RTS) | | 1.340 |

Source: Model results, (2020).

*** $p < 0.01$,

** $p < 0.05$,

* $p < 0.1$.

1.34 points. This implies that on average, dry-season vegetable farmers are operating at an increasing rate of return to scale in the Upper East region. In other words, when all the input costs are jointly increased by 1%, the output value of vegetables will increase by 1.34%.

Again, vegetable farms in the Upper East region are operating in the first stage of the production function. For this reason, farmers could take advantage of expanding their farms to enjoy the advantage of economies of scale. However, expanding vegetable farms will also be associated with increasing expenditure on the costs of irrigating the farms. The availability of water remains a challenge for vegetable production during the dry season, hence must be provided if vegetable production has to be expanded in the study area.

## 4.5 Technical efficiency estimates

The distribution of the technical efficiencies of dry season vegetable farms in the Upper East Region of Ghana is reported in Fig 1. The results depict that the distribution of the technical efficiencies ranges between a minimum and maximum technical efficiency score. The graph shows that 188 out of the 322 farmers recorded technical efficiency scores between 80% and below while 134 farmers obtained technical efficiency scores above 80% and 99%.

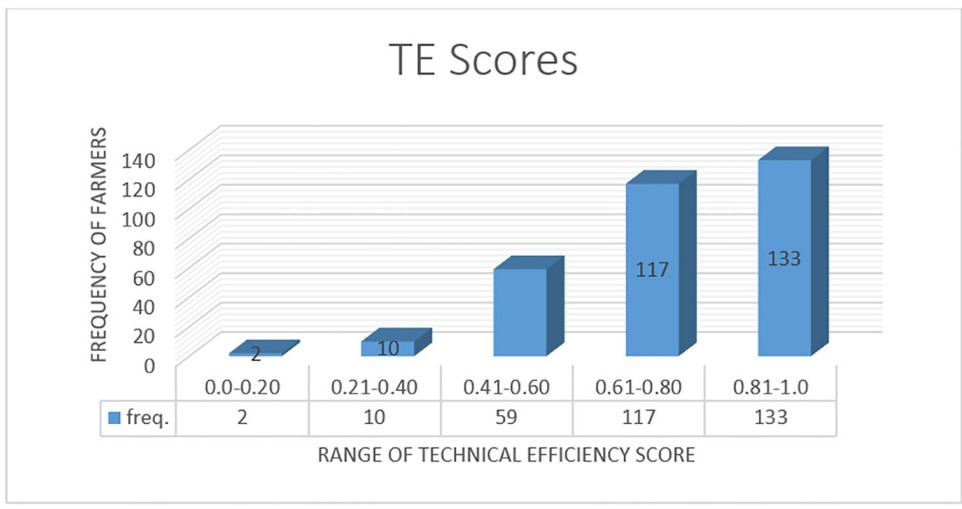

**Fig 1. Distribution of technical efficiency scores of vegetable farmers.**

The modal technical efficiency score shows that 133 observations operate within the efficiency scores of 81% and 100%. The overall predicted mean technical efficiency score is 0.73 points. This means that on average, dry-season vegetable farms produce about 73% of their potential (stochastic) frontier output value, given the current level of technology and input costs. The implication is that about 27% of the farms' potential output is unaccounted for due to technical inefficiency. This, therefore means that there is the possibility of increasing vegetable productivity in Ghana by an average of 27% within the shortest possible time with the same disposable resources by adopting the practices of the best farm.

## 4.6 Determinants of technical inefficiency of dry-season vegetable farmers

The results in Table 9 show the effects of the exogenous variables on the technical inefficiency of dry-season vegetable farmers. Most of the variables associated with technical inefficiency, had the expected signs, though some relationships were weak. The findings indicate that variables which had significant coefficients included household size, extension visits, use of motorized water pump system, use of gravity-fed irrigation system, and years of experience.

Contrary to the a priori expectation, the coefficient of household size is estimated to be positive and significant at a 1% level. This implies that farm families with relatively larger household sizes are relatively less efficient than those with smaller household sizes. [118], obtained

**Table 9. Sources of technical inefficiency of vegetable farmers.**

| Variable | Parameter | Coefficient | Standard error |
|---|---|---|---|
| Constant | $\delta_0$ | -2.212 | 1.503 |
| Age | $\delta_1$ | 0.018 | 0.060 |
| Age squared | $\delta_2$ | 0.000 | 0.001 |
| Gender | $\delta_3$ | 0.623 | 0.428 |
| Household size | $\delta_4$ | 0.0945*** | 0.033 |
| **Farmer's Edu. level** | | | |
| No education | | | |
| Basic education | $\delta_5$ | 0.118 | 0.280 |
| SHS/Tech/Voc. education | $\delta_6$ | 0.182 | 0.413 |
| Tertiary education | $\delta_7$ | -0.815 | 0.496 |
| Extension visits | $\delta_8$ | -0.117* | 0.068 |
| **Irrigation Technology** | | | |
| Motorized Pump | $\delta_9$ | -0.872** | 0.369 |
| Gravity-fed | $\delta_{10}$ | -1.415*** | 0.510 |
| Number of irrigations (watering) | $\delta_{11}$ | 0.001 | 0.004 |
| Experience | $\delta_{12}$ | -0.076*** | 0.019 |
| Credit access | $\delta_{13}$ | -0.180 | 0.319 |
| **Crop type** | | | |
| Pepper | $\delta_{14}$ | -0.398 | 0.285 |
| Onions | $\delta_{15}$ | -0.179 | 0.328 |
| Tomatoes | $\delta_{16}$ | 0.450 | 0.328 |
| Garden eggs | $\delta_{17}$ | -0.178 | 0.483 |

Source: Model results, (2020).

Note:

*** $p < 0.01$,

** $p < 0.05$,

* $p < 0.1$.

similar findings in their study of technical efficiency and its determinants of chilli farms in the Volta region of Ghana. A summary statistic of the data shows that about 61% of the sampled dry-season vegetable farms are less than 0.5ha. Hence, depending on the skills of the family labour deployed on the farm, too much labour input on a small vegetable farm size (some less than 0.5ha) could also lead to diminishing marginal returns on labour. Besides, the results also lend credence to some findings which argue that larger household sizes may not provide labour to improve technical efficiency since large household sizes may be composed of many children who are always in school [119].

Consistent with the results of [120], the number of extension visits positively influences technical efficiency at a 10% significant level. This implies that farmers who have access to regular extension education are more technically efficient than those with few or no access to extension education. This is premised on the fact that improved technologies, knowledge, skills and inputs developed by scientists get to farmers only via agricultural extension agents (AEAs). The implication is that farmers who have access to extension services and become technologically abreast can help improve yields and with good selling prices, will achieve high output returns (profit).

The coefficients of the motorized water pump ($p < 0.05$) and gravity-fed ($p < 0.01$) irrigation systems adopted by farmers were positive and significantly associated with the technical efficiency of dry season vegetable production. This means that farmers who use motorized water pumps or gravity-fed irrigation technology are more technically efficient relative to farmers who adopt the manual system of irrigation. The findings, however, have implications on the cost of irrigation as corroborated by [121]. The motorized pump, in particular, is the most common system adopted by the farmers for irrigating vegetable fields in the study area. However, its running cost has been a challenge to farmers, in terms of operation and maintenance (eg. purchase of fuel), thus, making irrigating vegetable fields very expensive for farmers in the study area (Personal communication, 2019).

The coefficient of experience had a positive and significant relationship with the technical efficiency of dry-season vegetable production. The implication is that vegetable farming depends on years of accumulated experience, which gives better managerial skills for high performance and higher yields. This finding concurs with the results of [24,109] but contradicts the finding of [121]. The analysis of the study was controlled by including the type of vegetable crops (pepper, onions, tomatoes, and garden eggs) cultivated in the analysis.

## 4.7 Summary of key findings of the study

This study examines the production risk and technical efficiency of dry-season vegetable farmers in the Upper East Region of Ghana. Specifically, the study determines the technical efficiency of dry-season vegetable farmers who produce pepper, onions, tomatoes, and garden eggs. The key findings of the study are as follows:

The study reveals that employing inputs such as labour, fertilizer, agrochemicals and improved seeds increases the output value of vegetables per hectare. The most important yield-boosting inputs in dry-season vegetable production are labour, followed by fertilizer and agrochemicals. The analysis of the production risk shows that costs associated with labour, seeds and agrochemicals increase the variability of the output value of vegetables while fertilizer and irrigations cost increase the variability of the output value of vegetables in the study location. The mean elasticities of 1.34 of vegetable production imply that on average, dry season vegetable farmers in the Upper East Region exhibit increasing return to scale hence, are operating in the first stage of the production function. Regarding the distribution of technical efficiencies, it was revealed that the farmers operate at a minimum technical efficiency of 7%

and a maximum technical efficiency of 99% with a mean technical efficiency score of 73%. In other words, the technical inefficiency level of the farmers is about 27%. To improve the technical efficiency of the vegetable farmers, issues relating to the number of extension visits, the use of motorised water pump systems and gravity-fed irrigation systems, as well as the experience level of the farmers need to be given critical attention in the study area.

## 5. Conclusion and recommendations of the study

Dry season vegetable production is a major economic activity for the people in northern Ghana, and the Upper East Region in particular but low productivity of the farms remains a drawback to the farmers, making them unable to generate the needed income to sustain their livelihoods. Inefficiency in the utilization of existing resources is the main challenge affecting productivity income and household food security of farmers. Drawing from the empirical findings, this study makes the following conclusions:

Dry-season vegetable farmers in the Upper East Region who are technically efficient have an average higher vegetable farm output which gives them higher income. Farmers who use more inputs like labour, fertilizer, agrochemicals and improved seeds obtain a higher output value of vegetables per hectare than those who use less. Investing in costs associated with labour, seed and agrochemicals decreases production risks thus increasing the output value of vegetables while costs attributed to fertilizer and irrigations increase production risks and decrease the output value of vegetables in the study location. The dry season farmers are operating at the first stage of the production process and hence are in an increasing return to scale. Therefore, to improve the production and productivity of vegetables in the area, the Irrigation Developing Authority (IDA) should rehabilitate the existing irrigation system in the region, supply dry-season farmers with more efficient water-saving irrigation equipment and kits (eg. water pumps, sprinklers, drip) at subsidised prices as part of the package under the One Village One Dam (1DV1D) flagship project to enable the farmers efficiently irrigate their fields to increase productivity and incomes. To reduce the risk and costs associated with the use of fertilizer, farmers should reduce the application of inorganic fertilizer, and instead increase the application of organic fertilizer to improve the productivity of farms and increase income. Farmers should devise efficient ways of applying fertilizer on the farms to minimize cost whilst maximizing the yield levels to increase the income of farmers. The existing fertilizer and seed subsidies by the Government should be expanded to include agrochemicals as well as support experienced farmers to mentor young and inexperienced farmers with credit to increase their scale of operations to increase vegetables' output and (profit) income.

## Supporting information

**S1 Data.**
(XLSX)

**S1 File.**
(DOCX)

## Acknowledgments

The study is an excerpt from the author's PhD thesis topic "Technical Efficiency of Dry Season Vegetable Farmers and Its Implications on Households' Food Security in the Upper East Region of Ghana". The authors are grateful to all respondents for participating in the study and providing consent to publish the research's findings after removing identifiable

information. We are also grateful to all examiners/reviewers of the study for their comments and contributions which helped in shaping the manuscript.

## Author Contributions

**Conceptualization:** James Anaba Akolgo, D. B. Sarpong.

**Data curation:** James Anaba Akolgo.

**Formal analysis:** James Anaba Akolgo.

**Methodology:** James Anaba Akolgo, Y. B. Osei-Asare, D. B. Sarpong.

**Resources:** James Anaba Akolgo.

**Supervision:** Y. B. Osei-Asare, D. B. Sarpong, Freda E. Asem, Wilhemina Quaye.

**Validation:** Y. B. Osei-Asare, D. B. Sarpong, Freda E. Asem, Wilhemina Quaye.

**Writing – original draft:** James Anaba Akolgo.

**Writing – review & editing:** James Anaba Akolgo, Freda E. Asem, Wilhemina Quaye.

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
