## [Decision Letter · Decision Letter 0]

11 Jan 2024

PONE-D-23-33204Production Risk and Technical Efficiency of Dry-Season Vegetable Farmers in the Upper East Region of Ghana.PLOS ONE

Dear Dr. Akolgo,

Thank you for submitting your manuscript to PLOS ONE. After careful consideration, we feel that it has merit but does not fully meet PLOS ONE’s publication criteria as it currently stands. Therefore, we invite you to submit a revised version of the manuscript that addresses the points raised during the review process.

**ACADEMIC EDITOR: I am pleased to inform you that two anonymous reviewers have reviewed your manuscript and you are expected to attend to their comments/suggestions as early as you can. Thank you.**==============================

Please include the following items when submitting your revised manuscript:A rebuttal letter that responds to each point raised by the academic editor and reviewer(s). You should upload this letter as a separate file labeled 'Response to Reviewers'.A marked-up copy of your manuscript that highlights changes made to the original version. You should upload this as a separate file labeled 'Revised Manuscript with Track Changes'.An unmarked version of your revised paper without tracked changes. You should upload this as a separate file labeled 'Manuscript'.

We look forward to receiving your revised manuscript.

Kind regards,

Olutosin Ademola Otekunrin

Academic Editor

PLOS ONE

A clean copy of the edited manuscript (uploaded as the new *manuscript* file).

4. You indicated that ethical approval was not necessary for your study. We understand that the framework for ethical oversight requirements for studies of this type may differ depending on the setting and we would appreciate some further clarification regarding your research. Could you please provide further details on why your study is exempt from the need for approval and confirmation from your institutional review board or research ethics committee (e.g., in the form of a letter or email correspondence) that ethics review was not necessary for this study? Please include a copy of the correspondence as an ""Other"" file.

5. We note that you have referenced (Effiong, E. (2005). Efficiency of production in selected livestock enterprises in Akwa Ibom State, Nigeria. Unpublished Ph. D Dissertation. Department of Agricultural Economics, Michael Okpara University of Agriculture, Umudike.) and (Idiong, I. (2006). Evaluation of technical, allocative and economic efficiencies in rice production systems in Cross River State, Nigeria. Unpublished Ph. D Dissertation. Michael Okpara University of Agriculture, Umudike.) which has currently not yet been accepted for publication. Please remove this from your References and amend this to state in the body of your manuscript: (ie “Bewick et al. [Unpublished]”) as detailed online in our guide for authors

6. We note that Figure 1 in your submission contain [map/satellite] images which may be copyrighted. All PLOS content is published under the Creative Commons Attribution License (CC BY 4.0), which means that the manuscript, images, and Supporting Information files will be freely available online, and any third party is permitted to access, download, copy, distribute, and use these materials in any way, even commercially, with proper attribution. For these reasons, we cannot publish previously copyrighted maps or satellite images created using proprietary data, such as Google software (Google Maps, Street View, and Earth). For more information, see our copyright guidelines: http://journals.plos.org/plosone/s/licenses-and-copyright.

Reviewers' comments:

Reviewer's Responses to Questions

Comments to the Author

1. Is the manuscript technically sound, and do the data support the conclusions?

Reviewer #1: Partly

Reviewer #2: Yes

2. Has the statistical analysis been performed appropriately and rigorously? 

Reviewer #1: Yes

Reviewer #2: Yes

3. Have the authors made all data underlying the findings in their manuscript fully available?

Reviewer #1: Yes

Reviewer #2: Yes

4. Is the manuscript presented in an intelligible fashion and written in standard English?

Reviewer #1: No

Reviewer #2: Yes

5. Review Comments to the Author

Reviewer #1: The study is quite interesting but need some major revisions for making it more understandable. There is high recommendation to recheck sentence structure and grammatical error. The paper should help by the number of lines that reviewers can provide suggestion through line numbers.

Besides, the following recommendations and revisions are needed:

Abstract:

1. “Farmers in the Upper East Region of Ghana have taken advantage of the increasing demand for vegetable in Ghana to intensify vegetable production during dry season to raise income but the outputs have been low.” What does it mean? Do the citizens aware about their health problems and want to take vegetables? There are several studies found that citizens of developing countries like Ghana are not aware of taking balanced diet.

Though, the authors mentioned that the production of vegetables are low but farmer can take advantages of producing vegetables by what? exporting vegetables? Or increasing vegetables intake?

2. “Furthermore, labour cost, seed cost and agrochemical costs decrease the variability of

vegetable output whilst fertilizer and irrigation costs increase the variability of returns of vegetable production.” What does “variability” mean here? To what extent the variability exposes?

Introduction:

1. The food insecurity and poverty are triggered by not only COVID-19 but also conflict and climate change. In fact, conflict and climate change triggers it more than COVID-19.

2. The motive of the study is not cleared here.

3. Justification of the study is also not cleared here.

4. A constructive literature review is highly recommended here.

The author would benefit from the following papers:

https://www.researchgate.net/publication/372685863_Climate_change_and_agriculture_nexus_in_Bangladesh_Evidence_from_ARDL_and_ECM_techniques

https://www.researchgate.net/publication/369013150_Economic_assessment_of_rice_farmers'_climate_change_adaptation_options_and_their_sustainability_a_case_of_Pabna_district_Bangladesh

https://www.researchgate.net/publication/356503914_Assessing_Livelihood_Adaptation_Indices_and_the_Sustainability_of_Rice_Farmers_in_Bangladesh's_Northwestern_Region

https://www.researchgate.net/publication/315368214_Perception_of_and_Adaptive_Capacities_to_Climate_Change_Adaptation_Strategies_and_Their_Effects_on_Rice_Production_A_Case_of_Pabna_District_Bangladesh

Methodology

1. The explanation of model is unclear about the estimation. The stochastic frontier model is used here for estimating technical efficiency but the technical efficiency here are not explain the adaptation of technology. That’s why here used production risk model.

But in result section, both models were estimated to explain both technical efficiency and production risk.

2. Why does elasticity estimation need here, there is no justification mentioned here.

3. There is no explanation of any model which mentioned the estimation of source of technical inefficiency.

Results:

1. In efficiency model both frontier model and average model were needed to estimate to compare which is best fit.

2. In explanation, how much the variables increase or decrease the efficiency, didn’t mention.

3. Which variables increase or decrease production risk mention here but didn’t mention how much increase or decrease production risk.

Summary:

Summary of key finding needs to revise again after revising the mentioned revisions.

Conclusion:

The conclusion and recommendations are nothing but summarise of the study. A revised conclusion and recommendations are recommended.

Reviewer #2: The study investigated production risk, technical efficiency, and the determinants of dry-season vegetable production in Ghana. The article is well-written and addresses a key issue. I only have minor concerns. Below are my suggestions and comments.

Add the aim of the study to the background of the abstract.

State the independent variables used in the inefficiency model.

In the result section, also give the monetary values in an international currency (USD) to enable readers to understand the value.

There is a need to further discuss the result, especially the maximum likelihood estimates of the trans-log mean stochastic function. Give possible reasons for each significant independent variable.

A summary of the study is not necessary as it is a repetition of what you have under condition.

6. PLOS authors have the option to publish the peer review history of their article (what does this mean?). If published, this will include your full peer review and any attached files.

Do you want your identity to be public for this peer review? For information about this choice, including consent withdrawal, please see our Privacy Policy.

Reviewer #1: No

Reviewer #2: Yes: Ridwan Mukaila

---

## [Author Response · Author response to Decision Letter 0]

12 Mar 2024

Response to reviewers is attached

---

## [Decision Letter · Decision Letter 1]

7 May 2024

PONE-D-23-33204R1Production Risk and Technical Efficiency of Dry-Season Vegetable Farmers in the Upper East Region of Ghana.PLOS ONE

Dear Dr. Akolgo,

Thank you for submitting your manuscript to PLOS ONE. After careful consideration, we feel that it has merit but does not fully meet PLOS ONE’s publication criteria as it currently stands. Therefore, we invite you to submit a revised version of the manuscript that addresses the points raised during the review process.

Editor's Comments:I am pleased to inform you that experts in the field have reviewed your manuscript. Kindly address the comments of reviewer 1 as early as possible. Thank you.

We look forward to receiving your revised manuscript.

Kind regards,

Olutosin Ademola Otekunrin

Academic Editor

PLOS ONE

Reviewers' comments:

Reviewer's Responses to Questions

**Comments to the Author**

1. If the authors have adequately addressed your comments raised in a previous round of review and you feel that this manuscript is now acceptable for publication, you may indicate that here to bypass the “Comments to the Author” section, enter your conflict of interest statement in the “Confidential to Editor” section, and submit your "Accept" recommendation.

Reviewer #1: (No Response)

Reviewer #2: All comments have been addressed

2. Is the manuscript technically sound, and do the data support the conclusions?

Reviewer #1: Partly

Reviewer #2: Yes

3. Has the statistical analysis been performed appropriately and rigorously? 

Reviewer #1: No

Reviewer #2: Yes

4. Have the authors made all data underlying the findings in their manuscript fully available?

Reviewer #1: Yes

Reviewer #2: Yes

5. Is the manuscript presented in an intelligible fashion and written in standard English?

Reviewer #1: No

Reviewer #2: Yes

6. Review Comments to the Author

Reviewer #1: The author didn’t address all the point of revision. It was recommended that the line number need to add. But author didn’t add any number line which mislead me in revision. The authors tried to improve the grammatical error. But still, they should check it again. There is major grammatical error in the whole writing.

Abstract: 1. “Farmers in the Upper East Region …….. abysmally low.” Please remove intensified.

2. “With the present state of technology and resource constraints of farmers……” What does it mean by options? Please clarify.

Introduction: 1. Still justification and motive of the study is still unclear.

Literature: A constructive literature was recommended. Author didn’t address any literature review.

The author would benefit from the following papers:

https://www.researchgate.net/publication/372685863_Climate_change_and_agriculture_nexus_in_Bangladesh_Evidence_from_ARDL_and_ECM_techniques

https://www.researchgate.net/publication/369013150_Economic_assessment_of_rice_farmers'_climate_change_adaptation_options_and_their_sustainability_a_case_of_Pabna_district_Bangladesh

https://www.researchgate.net/publication/356503914_Assessing_Livelihood_Adaptation_Indices_and_the_Sustainability_of_Rice_Farmers_in_Bangladesh's_Northwestern_Region

https://www.researchgate.net/publication/315368214_Perception_of_and_Adaptive_Capacities_to_Climate_Change_Adaptation_Strategies_and_Their_Effects_on_Rice_Production_A_Case_of_Pabna_District_Bangladesh

Methodology:

1. The authors use production risk function to capture farm’s risk behaviour. But why did you use only technical efficiency model? Isn’t both technical and profit efficiency would calculate here to understand farmer’s behaviour in risky situation?

Result:

1. In explanation, how much the variables increase or decrease the efficiency, didn’t mention yet.

2. Which variables increase or decrease production risk mention here but still didn’t mention how much increase or decrease production risk.

3. the authors didn’t link their result to previous study.

Limitation: What are the limitations of the study? Mention the limitations.

Reviewer #2: I do not have additional comments, as my earlier comments have been adequately addressed.

7. PLOS authors have the option to publish the peer review history of their article (what does this mean?). If published, this will include your full peer review and any attached files.

Reviewer #1: No

Reviewer #2: **Yes: **Ridwan Mukaila

---

## [Author Response · Author response to Decision Letter 1]

14 Jul 2024

B. REVIEWER 2 

ABSTRACT

1. The abstract has been revised to conform to the suggestions. Farmers in the Upper East Region …….. abysmally low.” Intensified has been deleted. 

2. Clarification on the statement “With the present state of technology and resource constraints of farmers……” has been clarified. 

INTRODUCTION 

The entire introduction has also been revised to: 

1. Explain the move for the study in the last paragraph.

2. Provide a constructive literature review

METHODOLOGY

1. Yes, the study indeed employed the production risks model incorporated into the technical efficiency model to capture the farm’s risk behavior. The profit efficiency model was not included in the calculation because farm-level profits were not calculated in the analysis. The value of output in monetary terms was an aggregation for the dependent variable since each farmer produces more than one vegetable in a season. This is consistent with the literature that states that “technical efficiency studies on the products of the whole farm, rather than specific products usually apply monetary value instead of physical quantities” [56, 95-99]. 

RESULTS

1. The input variables determine the elasticities of the production process. 

2. The socio-economic variables explain the inefficiencies of the production process where the direction of the signs (either + or -) indicates whether it increases or decreases the inefficiency.

3. Also, the positive (+) signs of the production risk variables tell whether the variable is risk increasing or negative (-) signs show risk decreasing. So, we use the sign direction of the variable to explain and not the absolute value of the variable.

4. The results section has been revised to link the explanation to the literature.

The limitations of the study

The data on the irrigation of the vegetables should have been in terms of the volume of water applied per day instead of the frequency of applications per season. This does not tell the exact quantity of water requirement per farm size (ha).

---

## [Decision Letter · Decision Letter 2]

12 Aug 2024

Production Risk and Technical Efficiency of Dry-Season Vegetable Farmers in the Upper East Region of Ghana.

PONE-D-23-33204R2

Dear Dr. Akolgo,

We’re pleased to inform you that your manuscript has been judged scientifically suitable for publication and will be formally accepted for publication once it meets all outstanding technical requirements.

Kind regards,

Olutosin Ademola Otekunrin

Academic Editor

PLOS ONE

Additional Editor Comments (optional):

Reviewers' comments:

Reviewer's Responses to Questions

**Comments to the Author**

1. If the authors have adequately addressed your comments raised in a previous round of review and you feel that this manuscript is now acceptable for publication, you may indicate that here to bypass the “Comments to the Author” section, enter your conflict of interest statement in the “Confidential to Editor” section, and submit your "Accept" recommendation.

Reviewer #1: All comments have been addressed

2. Is the manuscript technically sound, and do the data support the conclusions?

Reviewer #1: Yes

3. Has the statistical analysis been performed appropriately and rigorously? 

Reviewer #1: Yes

4. Have the authors made all data underlying the findings in their manuscript fully available?

Reviewer #1: Yes

5. Is the manuscript presented in an intelligible fashion and written in standard English?

Reviewer #1: Yes

6. Review Comments to the Author

Reviewer #1: (No Response)

7. PLOS authors have the option to publish the peer review history of their article (what does this mean?). If published, this will include your full peer review and any attached files.

Reviewer #1: **Yes: **Farjana Eyasmin

---

## [Editor Report · Acceptance letter]

17 Dec 2024

PONE-D-23-33204R2 

PLOS ONE

Dear Dr. Akolgo, 

I'm pleased to inform you that your manuscript has been deemed suitable for publication in PLOS ONE. Congratulations! Your manuscript is now being handed over to our production team.

Kind regards, 

on behalf of

Dr. Olutosin Ademola Otekunrin 

Academic Editor

PLOS ONE